# Blood Immunophenotypes of Idiopathic Pulmonary Fibrosis: Relationship with Disease Severity and Progression

**DOI:** 10.3390/ijms241813832

**Published:** 2023-09-07

**Authors:** Nuria Mendoza, Sandra Casas-Recasens, Núria Olvera, Fernanda Hernandez-Gonzalez, Tamara Cruz, Núria Albacar, Xavier Alsina-Restoy, Alejandro Frino-Garcia, Gemma López-Saiz, Lucas Robres, Mauricio Rojas, Alvar Agustí, Jacobo Sellarés, Rosa Faner

**Affiliations:** 1Institut d’Investigacions Biomediques August Pi i Sunyer (IDIBAPS), 08036 Barcelona, Spain; nmendoza@recerca.clinic.cat (N.M.); sacasas@recerca.clinic.cat (S.C.-R.); olvera@recerca.clinic.cat (N.O.); fhernandez@clinic.cat (F.H.-G.); cruz@recerca.clinic.cat (T.C.); albacar@clinic.cat (N.A.); aagusti@clinic.cat (A.A.); sellares@clinic.cat (J.S.); 2Centro de Investigación Biomédica en Red de Enfermedades Respiratorias (CIBERES), 28029 Madrid, Spain; lrobres@recerca.clinic.cat; 3Biomedicine Department, Universitat de Barcelona, 08036 Barcelona, Spain; 4Barcelona Supercomputing Center (BSC), 08034 Barcelona, Spain; 5Respiratory Institute, Clinic Barcelona, 08036 Barcelona, Spain; xalsina@clinic.cat (X.A.-R.); adfrino@clinic.cat (A.F.-G.); gsaiz@recerca.clinic.cat (G.L.-S.); 6Department of Internal Medicine, The Ohio State University Wexner Medical Center, Columbus, OH 43210, USA; mauricio.rojas@osumc.edu

**Keywords:** interstitial lung diseases, antifibrotic therapy, immunity and inflammation

## Abstract

(1) The role of the immune response in the pathogenesis of idiopathic pulmonary fibrosis (IPF) remains controversial. We hypothesized that peripheral blood immune phenotypes will be different in IPF patients and may relate to the disease severity and progression. (2) Whole blood flow cytometry staining was performed at diagnosis in 32 IPF patients, and in 32 age- and smoking-matched healthy controls. Thirty-one IPF patients were followed up for one year and categorized as stable or progressors based on lung function, deterioration and/or death. At 18–60 months, immunophenotypes were characterized again. (3) The main results showed that: (1) compared to matched controls, at diagnosis, patients with IPF showed more neutrophils, CD8^+^HLA-DR^+^ and CD8^+^CD28^−^ T cells, and fewer B lymphocytes and naïve T cells; (2) in IPF, circulating neutrophils, eosinophils and naïve T cells were associated with lung function abnormalities; (3) patients whose disease progressed during the 12 months of follow-up showed evidence of cytotoxic dysregulation, with increased CD8^+^CD28^−^ T cells, decreased naïve T cells and an inverted CD4/CD8 ratio at baseline; and (4) blood cell alterations were stable over time in survivors. (4) IPF is associated with abnormalities in circulating immune cells, particularly in the cytotoxic cell domain. Patients with progressive IPF, despite antifibrotic therapy, present an over-activated and exhausted immunophenotype at diagnosis, which is maintained over time.

## 1. Introduction

Idiopathic pulmonary fibrosis (IPF) is a progressive lung disease characterized by the accumulation of scar tissue, subepithelial fibroblast foci and microscopic honeycombing [1]. Although the pathogenesis of IPF remains incompletely understood, it is thought that abnormally activated and/or damaged lung epithelial cells secrete a panel of mediators that differentiate fibroblasts to myofibroblasts which, in turn, deposit large amounts of extracellular matrix that destroys the normal lung architecture and contribute to the recruitment of inflammatory cells [2,3]. The original triggers causing the abnormal activation of lung epithelial cells are varied and include cigarette smoking, chronic viral infections, accelerated aging and genetic predisposition [4]. It is generally believed that these repeated micro injuries, in combination with exaggerated wound repair and dysregulated tissue remodelling, lead to IPF [5] with impaired lung function, breathlessness and, eventually, death [6,7].

The role of the immune response in the pathogenesis of IPF remains controversial. On the one hand, abnormal innate and adaptive immune responses have been identified in the lung tissue of patients with end stage IPF [2], including changes in the amount of the infiltrate and/or functionality of neutrophils, macrophages and natural killer cells (NKs) [2], and in this setting they are considered as an epiphenomenon of the fibrotic process. Likewise, in peripheral blood, the decreased expression of T cell regulatory genes [8], downregulation of the T cell costimulatory molecule CD28 expression (a marker of lymphocyte exhaustion) [9,10] and increased expression of CXCL13 (a B cell lymphoid follicle homing cytokine) [11] have also been described in patients with IPF. On the other hand, however, IPF patients treated with immunosuppressors are at increased risk of death and hospitalization [12]. We reasoned that a detailed characterization of the blood immune cell phenotype of IPF patients at diagnosis can provide new insights into the pathogenesis and progression of IPF over time. To explore this hypothesis, this study sought to: (1) compare the immune-phenotypes in the peripheral blood of patients with IPF at the time of diagnosis vs. healthy controls; (2) explore their potential associations with the severity of the disease at baseline and with disease progression during a 1-year follow-up; and (3) explore if these associations were maintained long-term, during 18–60 months of follow-up in survivors.

## 2. Results 

### 2.1. Participant Characteristics

Table 1 presents the main demographic and clinical characteristics of patients and controls at recruitment and in patients during a 1-year follow-up. By design (matching procedure), at recruitment, age (about 71 years) and smoking history were similar. As expected, lung function was abnormal in patients and normal in controls. At recruitment, 12 patients (37.5%) were receiving antifibrotic treatment and 20 (62%) were not (Table 1). A total of 31 of the 32 patients profiled at recruitment (96%) were followed-up for 1 year (Table 1, Appendix A). Disease progression, defined as above (annual FVC decline ≥ 10%, DLCO decline ≥ 15% and/or death within the first-year of follow-up), occurred in 18 patients (58%), despite the fact that 16 of them (88.9%) received antifibrotic treatment (pirfenidone or nintedanib) during follow-up. At recruitment, age, gender, and smoking status was similar between stable and progressor IPF patients (Table 1). Lung function was different in both groups during the 1-year follow-up (Table 1). Seven progressors died during the follow-up (38.9%).

### 2.2. Blood Immunophenotype Differences in IPF Patients vs. Healthy Individuals at Baseline

We found significant differences in both the innate and adaptive immunophenotypes in the blood of IPF patients vs. healthy controls at baseline (Figure 1 and Appendix A). Specifically, the percentage of circulating neutrophils was higher, and that of B cells was lower, in IPF patients (Figure 1a). Also, the proportion of CD8^+^HLA-DR^+^ (i.e., cytotoxic activated cells) and CD8^+^CD28^−^ T cells (i.e., cytotoxic exhausted, mainly of effector and memory effector phenotype) was higher in IPF patients than in controls, whereas that of naive CD8^+^ T cells was lower (Figure 1b and Appendix A). Given that the later alterations resemble the immunophenotype in aged individuals, we performed a two-way ANOVA to evaluate the effect of age, and the differences in CD8^+^CD28^−^ and naïve CD8^+^ T cells remained statistically significant (*p* = 0.027 and *p* = 0.007, respectively). No differences were observed in the percentage of PD-1^+^ T cells. Th17 lymphocytes were also decreased in IPF while Th1 and Tregs (trend, *p* = 0.072) were increased, and, accordingly, the Th1/Th17 and Th17/Tregs ratios were different between patients and controls. (Figure 1c,d and Figure 2 and Appendix A).

### 2.3. Association between Blood Immunophenotype and Severity of IPF at Baseline

Figure 3a presents the correlation between the immune cell types and lung function at baseline in IPF patients. FVC was positively related to naïve CD4^+^ T cells and negatively related to central memory CD4^+^ T cells. DLCO was positively related to the proportion of circulating eosinophils and lymphocytes, and negatively related to neutrophils, the neutrophile-to-lymphocyte Ratio (NLR) and the monocyte-to-lymphocyte Ratio (MLR) (Appendix A).

### 2.4. Relationships between Baseline Blood Immunophenotype and Disease Progression

Figure 3b (and Appendix A) shows that, compared to stable patients, at baseline the percentage of circulating NKT-like cells, CD8^+^ T cells, effector memory CD4^+^, and CD8^+^CD28^−^ T cells was higher, and that of naive CD4^+^ and CD8^+^ T cells and central memory CD8^+^ T cells lower, in progressors. Accordingly, the CD4/CD8 ratio was lower in progressors, suggesting that a predominant T cell response to intracellular antigens (CD8^+^) [13] is associated with worse prognosis despite the use of antifibrotic treatment. Again, no statistically significant differences in the percentage of PD-1+ T cells were observed between those who were stable and progressors; a two-way ANOVA was used to evaluate the effect of age on CD8^+^CD28^−^ T cells and naïve CD4^+^ and CD8^+^ T cells, which was still significant (*p* = 6.67 × 10^−5^, *p* = 0.003, *p* = 0.017, respectively). Appendix A provides the differences between the three groups: controls, those who were stable, and progressors, showing the latter having one the greatest differences.

### 2.5. Machine Learning Classification

We used a machine learning method (Elastic Net [14]) to investigate what combination of immune cells, determined at baseline, maximized the classification between progressors and stable patients. The constructed model showed that the combination of T CD8^+^CD28^−^, memory effector CD4^+^ T cells, CD4/CD8 ratio, and NKT-like cells offered the best classification model of stable vs. progressor patients (Figure 3b, AUC of 0.94 and accuracy of 0.86).

### 2.6. Reproducibility of IPF Immunophenotypes at Long-Term

To assess the stability and reproducibility of the immune differences observed at baseline between stable and progressors, we determined the immune profile again in alive patients (stable or progressor) during 18 to 60 months of follow-up. Figure 4a,b presents the paired analysis between the baseline and the long term assessment, and shows that the populations included in the machine learning classification model, CD8^+^CD28^−^, memory effector CD4^+^ T cells, CD4/CD8 ratio, and NKT-like cells, were stable in the long-term.

## 3. Discussion

The three main novel observations of this study are that: (1) abnormalities in both the innate and adaptive immune response can be detected in the circulating blood of patients with IPF at diagnosis; (2) some of these abnormalities relate to the severity of lung function at diagnosis; and, (3) a specific and longitudinally stable immune phenotype characterized by increased NKT-like cells, CD8^+^ T cells with an exhausted phenotype, and less naïve T cells, with an impaired CD4/CD8 ratio, is associated with IPF progression over time (AUC 0.94), despite the use of antifibrotic treatment. Collectively, these observations contribute to a better understanding of the role of the immune response in IPF, provide new prognostic biomarkers of potential utility in clinical practice, and pinpoint novel potential therapeutic targets that deserve further research.

### 3.1. Previous Studies and Interpretation of the Findings

Neutrophils are key players in the acute phase of inflammation but, chronically, they can lead to tissue remodelling and fibrosis [15]. Previous studies in IPF have shown that neutrophil attraction and activation markers, such as IL-8 (interleukin-8) and G-CSF (granulocyte colony-stimulating factor), are increased in broncho-alveolar lavage fluid (BALF) and sputum, and that they may predict future exacerbations in IPF [16]. In blood, it has been seen that neutrophils and NLR can be an indicator of disease progression in IPF and other fibrosing interstitial lung diseases [17,18]. Our results are in keeping with these previous observations, adding, however, that they can already be observed in peripheral blood at the time of IPF diagnosis, particularly in patients with lower DLCO.

Some previous publications have suggested that BALF eosinophilia can be a marker of disease progression in IPF [19,20]. Others showed that higher peripheral eosinophil counts were associated with reduced lung function (FVC and DLCO), although they were not associated with disease progression, exacerbations or antifibrotic discontinuation [21]. In our study, we found that circulating eosinophils are positively correlated with DLCO% ref. at recruitment (Figure 3a, Appendix A) but did not correlate with disease progression during follow-up (Appendix A).

Previous studies reported that peripheral blood T cells in patients with well-established IPF present a surface signature characterized by the loss of co-stimulatory molecules, specifically CD28 [10]; our results at diagnosis (hence, the earlier stages of disease) agree with these previous findings. T CD28^−^ cells are antigen-experienced memory T cells that accumulate in multiple diseases [22]. Considered “exhausted” cells, they have short telomeres, express markers of senescence, and secrete high levels of perforin, granzymes, IFNγ, and TNFα [10]. Previous reports have shown an abundance of CD8^+^CD28^−^ T cells in explanted IPF lung tissues (end stage disease) and reported that they predict poor prognosis [9,10,22]. Our results complement these findings, showing that at diagnosis in blood, the percentage of CD8^+^CD28^−^ T cells is associated with a poor prognosis and lung function decline (both FVC and DLCO). On the other hand, it is well-known that the balance between activating and survival markers is critical for the homeostatic maintenance of T cell responses [23]. In this setting, an increase in T CD8^+^CD28^−^ might reflect a continuous increase in the effector T cell subset, causing a state of activated but inefficient immunological responses that alters the distribution of the memory CD8^+^ T cell population which, in turn, may contribute to the ongoing fibrotic scenario by a deficient clearance of intracellular antigens [24]. This hypothesis is in keeping with the clinical observation that the pharmacological inhibition of the immune response is associated with worse prognosis, and our observation that progressors present abnormalities in the CD4/CD8 ratio at recruitment that are maintained at follow-up. The normal ratio of CD4/CD8 is ≥1.8 [25], and an inverted ratio (i.e., <1) has traditionally been associated with immune senescence, myelodysplasia, and persistent viral infections such as HIV (Human immunodeficiency virus), HCMV (Human cytomegalovirus), and EBV (Epstein–Barr virus) [25,26,27]. Some previous studies have reported a higher prevalence of seropositive EBV, HCV (Hepatitis C virus), and HCMV in patients with IPF, suggesting that viral infections may play a role, either as agents that predispose and/or aggravate lung fibrosis [28]. Further studies are required to unveil if the altered CD4/CD8 ratio that we report in progressive IPF is associated with persistent viral infections.

Concomitant with the CD8 decrease, we observed an increase in NKT-like cells in progressors. These cells are instrumental in the response to infections, tumours, and autoimmune diseases [29]. Interestingly, although this population slightly increased over time in stable patients, this increase was smaller than that observed in progressors.

Still, little is known about the T cells phenotype in circulating blood in patients with IPF at diagnosis and its evolution with time. Here, we observed that, at diagnosis, IPF patients have reduced Th17 lymphocytes and increased Th1, leading to disrupted Th1/Th17 and Th17/Tregs ratios. A depletion of circulating Th17 cells, along with a non-compromised regulatory T cells (Treg) compartment (similarly to what is observed in cancer) has been previously described by Galati et al. in the blood of IPF patients [30]. Here, we observed a tendency towards increased Treg in IPF, approaching statistical significance (*p* = 0.07).

Finally, a reduction in the proportion of naive CD4^+^/CD8^+^ T cells and T cell repertoire have also been reported in patients with IPF in relation to immune-senescence which, in turn, relates to impaired virus-specific T cell responses [31]. In this setting, we found a positive correlation between circulating naive CD4^+^ T cells and baseline FVC, and reduced levels of both circulating naive CD4^+^ and CD8^+^ T cells in IPF progressors at both recruitment and a trend in the follow-up, despite the use of anti-fibrotic treatment.

In light of our findings, we hypothesize that an inefficient immune response may allow the disease to progress. However, additional studies are needed to explore if the immunophenotype is already altered in patients with minimal interstitial changes before the disease onset, if the immunophenotype might be different in progressors with or without acute exacerbations, and if our AI model could be of use in this setting to predict such outcomes.

### 3.2. Clinical Implications

The advent of new anti-fibrotic treatments improved the prognosis of patients with IPF [32] but, for reasons still unclear, some patients progress despite their use [33]. Our results here show that an exhausted cytotoxic blood immune profile is associated with disease progression despite the use of anti-fibrotic treatment (Figure 3b). These observations have two important clinical implications. First, they highlight potential targets for new studies and therapies directed to this exhausted phenotype. And second, though it needs to be confirmed in larger studies, they provide potentially relevant prognostic information for the practicing physician.

### 3.3. Strengths and Limitations

The main strength of our study is that it characterizes in detail a wide range of immune cell types in circulating blood, an easily accessible tissue in clinical practice or clinical research, at the time of IPF diagnosis. Likewise, the fact that these patients were followed up over time allowed us to investigate the relationship of these baseline immune-phenotypes with disease progression. Among the potential limitations, we acknowledge that our sample size is relatively small, and that there were more females among the controls, but were matched by two major confounders in cell populations, age and smoking [34,35]. Likewise, our results need validation in larger, likely multicentre, cohorts to confirm the predictive power of our classifying model.

## 4. Materials and Methods

### 4.1. Study Design and Ethics

This was a prospective, observational, and controlled study that enrolled 32 consecutive patients who were diagnosed with IPF in the Pneumology Service of Clinic Barcelona between July 2016 and July 2021, and 32 age- and smoking-matched healthy individuals. Most IPF patients *n* = 26 (81.3%) were recruited at the time of diagnosis (some were referred from other centres where the diagnosis of IPF was already established and/or treatment initiated). Thirty one of them (96%) were followed up for 60 months (Appendix A). A first clinical visit at 12 months was used to determine disease stability or progression (see below). Survivors (*n* = 15) were visited again at 18–60 months. Biological samples were collected at recruitment and at 18–60 months. The diagnosis of IPF was established according to current international recommendations (ATS/ERS/JRS/ALAT) by a multidisciplinary committee [36]. All patients were treated according to international guidelines [37]. The Ethics Committee of our institution approved the study (HCB/2017/0901, 23 November 2017; HCB/2019/0687, 11 July 2019) and all participants signed their informed consent.

### 4.2. Measurements

#### 4.2.1. Lung Function and Disease Progression

Forced spirometry and the single breath carbon monoxide diffusing capacity of the lungs (DLCO) were measured according to international standards. Reference values were those of Roca et al. [38,39]. Disease progression was defined as an annual decline of relative forced vital capacity (FVC) ≥ 10%, DLCO ≥ 15% and/or death in the first year.

#### 4.2.2. Fluorescence-Activated Cell Staining Analysis (FACS)

We used FACS to profile B cells, T cells (and subpopulations), NK cells, NKT cells, monocytes, neutrophils, and eosinophils (Appendix A). Briefly, 120 μL of blood was incubated with 30 μL of the antibody mix (Appendix A) for 30 min at 4 °C. Then, erythrocytes were lysed (BD FACS Lysing Solution, San Jose, CA, USA) and cells were incubated with the Fixable Viability Stain (440UV, BD 566332) for 15 min at room temperature in the dark, washed, and fixed using PFA 4%. Fixed samples data were acquired with a LSRFortessa SORP (BD, San Jose, CA, USA). FlowJo version 10 software (FlowJo LL, Ashland, OR, USA) was used for analysis. The gating strategy to determine the populations is described in Appendix A.

### 4.3. Data Analysis

Results are presented as number, percentage, mean ± SD or median [95% CI]. The normality of the distribution of immune cell populations was tested with the Shapiro–Wilkinson test, and groups were compared via *t*-test or Mann–Whitney test, as appropriate. Correlations between immune cell populations and clinical parameters were tested using the Spearman test and considered significant if Rho > |0.3| and the *p* value < 0.05. All statistics were computed with R version 3.6.2 (12 December 2019) using custom scripts.

## 5. Conclusions

Patients with IPF show significant alterations in the peripheral blood immunological profile at diagnosis, both in their innate and adaptive immune responses, particularly in the cytotoxic compartment. Moreover, IPF progression is associated with T CD8^+^ cells dysregulation, and an inverted CD4/CD8 ratio suggesting an over-activated, aged, and “exhausted” immune status potentially related to intracellular antigens.

## Figures and Tables

**Figure 1 ijms-24-13832-f001:**
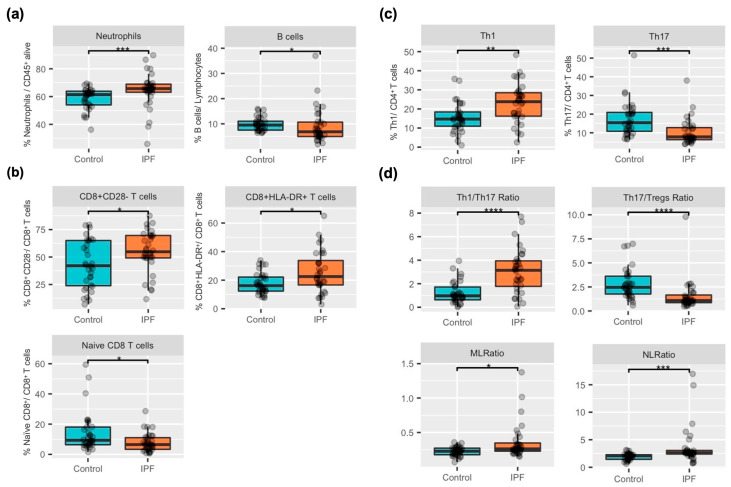
Characterization of the innate and adaptive compartment in IPF patients (orange box) and controls (blue box) (median [IQR]) (**a**) Comparison of % neutrophils and B lymphocytes. (**b**) Comparison of % CD8^+^CD28^−^, HLADR+, and naive CD8^+^ T cells from CD8^+^ pool. (**c**) Comparison of % Th1 and Th17 from CD4^+^ pool. (**d**) Comparison of Th1/Th17, Th17/Tregs, Neutrophil-to-lymphocyte (NRL), and Monocyte-to-lymphocyte (MLR) ratios. * *p* < 0.05, ** *p* < 0.01, *** *p* < 0.001 and **** *p* < 0.0001.

**Figure 2 ijms-24-13832-f002:**
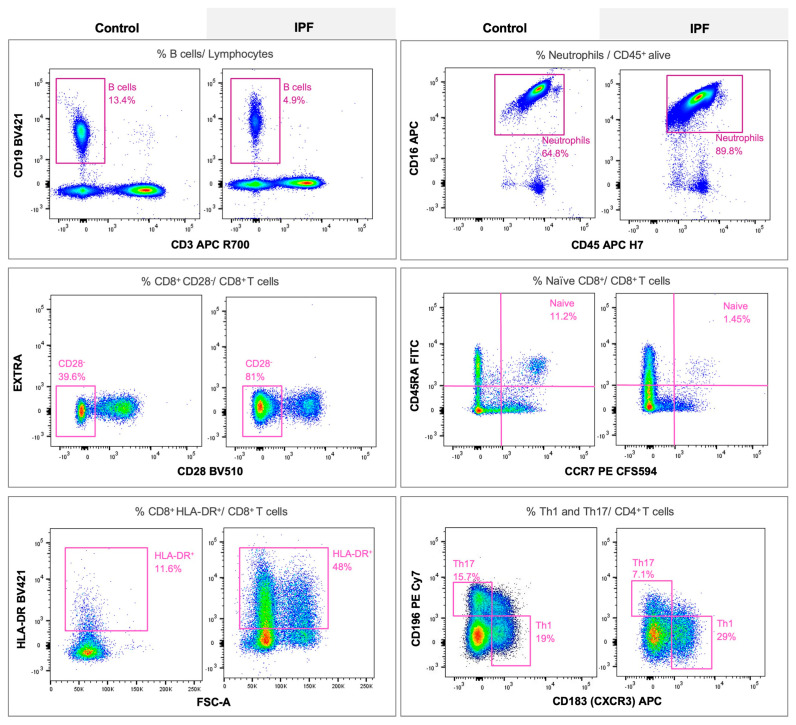
Representative flow cytometry gatings of controls and IPF patients of the peripheral immune populations that were statistically significant between the study groups: comparison of neutrophils and B lymphocytes, CD8^+^CD28^−^, CD8^+^HLADR^+^ and naive CD8^+^ T cells from CD8^+^ pool, and Th1 and Th17 from CD4^+^ pool.

**Figure 3 ijms-24-13832-f003:**
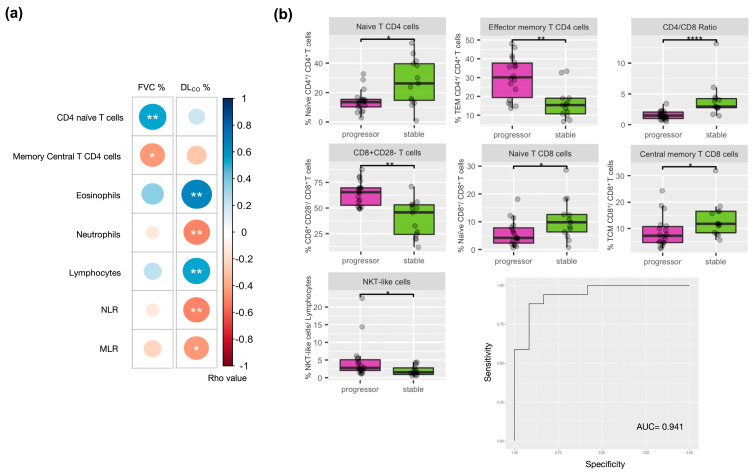
(**a**) Significant Spearman correlations between blood immune populations/ratios and lung function at recruitment in IPF patients. * *p* < 0.05, ** *p* < 0.01 and **** *p* < 0.0001. (**b**) Differential distribution of the baseline immune population in IPF progressors (pink box) and stable (green box); (median [IQR]): Comparison of % NKT-like cells; CD8^+^CD28^−^ from CD8^+^ pool, different CD8^+^ and CD4^+^ T memory cells and CD4/CD8 ratio; Receiving Operating Characteristics (ROC) curve of the Elastic Net multivariate analysis identifying the immune cell populations associated with disease progression.

**Figure 4 ijms-24-13832-f004:**
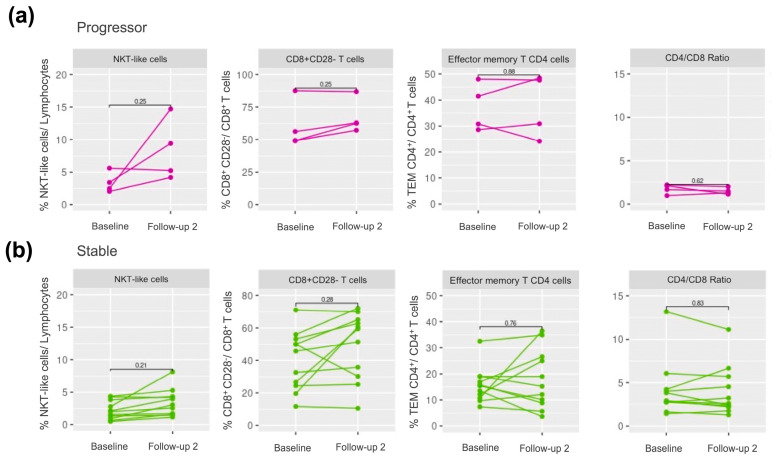
Paired analysis comparing the stability of the immune cell populations included in the Elastic Net, at baseline and second follow-up (18–60 months) in (**a**) progressors and (**b**) stable patients.

**Table 1 ijms-24-13832-t001:** Characteristics (*n* (%) or mean ± SD) of participants at study entry and during 12-month follow-up.

	At Study Entry			IPF Patients during 1-Year Follow-Up		
	Control (*n* = 32)	IPF (*n* = 32)	*p*-Value	Progressor (*n* = 18)	Stable (*n* = 13)	*p*-Value
Age	71.1 ± 5.17	71.6 ± 7.01	0.344	71.8 ± 5.99	70.7 ± 8.39	0.679
Males, *n* (%)	13 (40.6%)	23 (71.9%)	0.023	14 (77.8%)	9 (69.2%)	0.689
Smoking status			0.445			0.784
Former smoker	21 (65.6%)	25 (78.1%)		15 (83.3%)	10 (76.9%)	
Never smoker	6 (18.8%)	5 (15.6%)		3 (16.7%)	2 (15.4%)	
Current smoker	5 (15.6%)	2 (6.25%)		0 (0.00%)	1 (7.69%)	
BMI, Kg/m^2^	25.3 (3.5)	29.2 (8.72)	0.037	29.7 (10.0)	28.5 (6.81)	0.721
FVC, % ref.	106 (20.9)	69.8 (18.3)	<0.001	60.3 (10.8)	79.5 (16.8)	0.002
FEV1, % ref.	96.4 (17.2)	77.0 (16.7)	<0.001	69.3 (11.7)	85.3 (16.8)	0.008
FEV1/FVC, %	98.0 (8.27)	81.1 (5.80)	<0.001	83.1 (5.38)	78.9 (5.63)	0.055
DLCO, % ref.	NA	46.9 (16.8)		41.9 (14.7)	53.9 (17.9)	0.063
Antifibrotic before *, *n* (%)			<0.001			1.000
Yes	0 (0%)	12 (37.5%)		7 (38.9%)	5 (38.5%)	
Antifibrotic after *, *n* (%)			<0.001			0.497
Yes	0 (0%)	30 (93.8%)		16 (88.9%)	13 (100%)	
Antifibrotic drug, *n* (%)			<0.001			0.348
Nintedanib	0 (0%)	22 (73.3%)		13 (72.2%)	8 (61.5%)	
Pirfenidone	0 (0%)	8 (26.7%)		3 (16.7%)	5 (38.5%)	
Death, *n* (%)			1.000			0.025
Yes	0 (0%)	0 (0%)		7 (38.9%)	0 (0%)	

NA = Not available information; BMI = Body mass index; FVC = Forced Vital Capacity; FEV1 = Forced expiratory capacity 1 s; DLCO = Single breath carbon monoxide diffusing capacity; Antifibrotic before * = Antifibrotic treatment before recruitment; Antifibrotic after * = Antifibrotic treatment after recruitment.

## Data Availability

All data generated or analysed during this study are included in this published article [and its Appendix A].

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
