# Peer review of "Blood Immunophenotypes of Idiopathic Pulmonary Fibrosis: Relationship with Disease Severity and Progression"

_ijms, 2023, doi:10.3390/ijms241813832_

Round 1
Reviewer 1 Report
A much needed study characterising the immune cell phenotype of the peripheral blood of IPF patients at the moment of diagnosis and follow-up. Although many of the phenotypic changes have been previously described in IPF patients and/or associated with disease progression, this is the first study showing such changes are present as early as diagnosis. This paper, although limited in number of patients, opens up multiple possibilities to diagnose and predict the disease course in IPF patients. I particularly liked the AI model proposed, although a much larger number of patients will be needed for its validation. The paper is well written, the data is well presented and analysed. The only suggestion I would have would be to include the flow cytometry panels as main figures in the manuscript and show a few representative health vs IPF panels panels as examples for the quantified data. The discussion section was well written. In that regard, it would be very interesting to discuss if the phenotypic changes identified are present in patients with minimal interstitial changes, before the disease onset, and if in this population the AI model can also predict the progression to IPF. The limitations of the study have been well identified and discussed by the authors. It is very true that the limited numbers of patients at follow up is the biggest concern regarding the conclusions, but the data opens up many possibilities for future important translational research.
Author Response
POINT-BY-POINT RESPONSE
Manuscript: Blood Immunophenotypes of Idiopathic Pulmonary Fibrosis: Relationship with Disease Severity and Progression
Manuscript ID: ijms-2550704
Type of manuscript: Article
Review Comments to the Author
Reviewer # 1:
A much needed study characterizing the immune cell phenotype of the peripheral blood of IPF patients at the moment of diagnosis and follow-up. Although many of the phenotypic changes have been previously described in IPF patients and/or associated with disease progression, this is the first study showing such changes are present as early as diagnosis. This paper, although limited in number of patients, opens up multiple possibilities to diagnose and predict the disease course in IPF patients. I particularly liked the AI model proposed, although a much larger number of patients will be needed for its validation. The paper is well written, the data is well presented and analyzed.
We thank Reviewer 1 for her/his appreciation and assessment of our work as well as for the insightful and constructive comments made, which we address specifically below.
C1. The only suggestion I would have would be to include the flow cytometry panels as main figures in the manuscript and show a few representative health vs IPF panels as examples for the quantified data.
R1. We appreciate this comment. To address it we have added a new figure (Figure 2, page 4, lines 102-106) that shows representative Control vs. IPF flow cytometry panels. However, we would like to keep the two flow cytometry gating strategy figures in the supplement to ease the reading of the manuscript.
C2. The discussion section was well written. In that regard, it would be very interesting to discuss if the phenotypic changes identified are present in patients with minimal interstitial changes, before the disease onset, and if in this population the AI model can also predict the progression to IPF.
R2. Thank you for this comment, which we have incorporated in the revised version of the manuscript (page 7, lines 226-231). Unfortunately, however, we do not have the data needed to test this hypothesis right now, that should be addressed in future studies.
C3. The limitations of the study have been well identified and discussed by the authors. It is very true that the limited numbers of patients at follow up is the biggest concern regarding the conclusions, but the data opens up many possibilities for future important translational research.
R3. Thank you for this comment. We concur with it and hope that this pilot study encourages new and larger investigations.

Reviewer 2 Report
The authors analyzed immunophenotypes of patients with IPF and found altered immunophenotypes in IPF that were apparent in progressive IPF when compared to non-progressive IPF. Moreover, the altered phenotypes were stable during the observation period. The findings are interesting, and the manuscript was written well. However, there were several concerns about this paper.
1. The alternation of immunophenotypes in IPF (decreased naïve T cells and increased memory effector cells such as CD28- CD8+ T cells) resembles phenotypes in the aged subjects. Were the differences between groups significant when the age was adjusted as a covariant, although controls in this study were aged subjects?
2. In Fig.2b, it would be better to show the data of controls as a reference. Were the differences found between the sable IPF group and controls?
3. It is interesting and clinically important that the immunophenotypes could predict the progression of IPF, and immunophenotypes of the patients were unchanged during the observation period. Do the authors think that once the immune abnormality is established, the progression of ILD is determined? It would be better to discuss this point in more detail.
4. Were there patients who developed an acute exacerbation of IPF? If so, did these patients have characteristic immunophenotypes?
5. Please describe how ILD patients were selected. Were the participants consecutive patients who visited the authors’ hospital?
6. The authors showed the proportion of interested subsets. How about the numbers of the subsets? As clinical information, WBC numbers should be described.
7. CD56+ T cells might be better than NKT-like cells.
Author Response
POINT-BY-POINT RESPONSE
Manuscript: Blood Immunophenotypes of Idiopathic Pulmonary Fibrosis: Relationship with Disease Severity and Progression
Manuscript ID: ijms-2550704
Type of manuscript: Article
Review Comments to the Author
Reviewer #2
The authors analyzed immunophenotypes of patients with IPF and found altered immunophenotypes in IPF that were apparent in progressive IPF when compared to non-progressive IPF. Moreover, the altered phenotypes were stable during the observation period. The findings are interesting, and the manuscript was written well. However, there were several concerns about this paper.
We thank Reviewer 2 for her/his assessment of our work as well as for the insightful and constructive comment made, which we address specifically below.
C1. The alternation of immunophenotypes in IPF (decreased naïve T cells and increased memory effector cells such as CD28- CD8+ T cells) resembles phenotypes in the aged subjects. Were the differences between groups significant when the age was adjusted as a covariant, although controls in this study were aged subjects?
R1. Thank you for this comment. To rule out a potential aging effect on the immunophenotypes we used age and smoking matched healthy controls. Further, in the revised version, we performed a two-way ANOVA to evaluate the effect of age and that of the disease. Doing so, both in the comparison between stable IPF patients and progressors, and in the case control, although the age had an effect, the CD28-CD8+ T cells (p= 6.67e-05 and p=0.027, respectively) and naïve CD8+ T cells (p=0.017, p= 0.007, respectively), differences remained statistically significant. We now added this information to the revised version of our manuscript (see page 3, lines 89-92 and page 5, lines 129-131).
C2. In Fig.2b, it would be better to show the data of controls as a reference. Were the differences found between the sable IPF group and controls?
R2. We appreciate the proposal, however, we hold a differing viewpoint. The proposal aimed to identify the characteristics of progressors, which may not be essentially distinct from controls. Consequently, we believe that incorporating controls here might not yield supplementary insights. Nonetheless, in accordance with your suggestion, we have included a new table (Table S6) illustrating the disparities among the three groups. These distinctions are additionally elaborated upon in the main manuscript (page 5, lines 131-133, and page 9, lines 318-320).
C3. It is interesting and clinically important that the immunophenotypes could predict the progression of IPF, and immunophenotypes of the patients were unchanged during the observation period. Do the authors think that once the immune abnormality is established, the progression of ILD is determined? It would be better to discuss this point in more detail.
R3. Thanks for this very insightful comment, which we now discuss on page 7, lines 226-231. Indeed, although additional studies are needed to confirm this possibility, we indeed hypothesize that an inefficient immune response may underlie disease progression.
C4. Were there patients who developed an acute exacerbation of IPF? If so, did these patients have characteristic immunophenotypes?
R4. Thanks again for this very interesting comment. During follow-up, few progressors (n=5) presented an episode of acute exacerbation, and died. Unfortunately, this small number of individuals (5 out 18) is not statistically powered to explore if at baseline they present a characteristic immunophenotype, but we now add a comment on this in the discussion section (page 7, lines 226-231).
C5. Please describe how ILD patients were selected. Were the participants consecutive patients who visited the authors’ hospital?
R5. Thanks for this insight. As requested, we now clarify this in the revised version of our manuscript: “This was a prospective, observational, and controlled study that enrolled 32 consecutive patients who were diagnosed of IPF in the Pneumology Service of Clinic Barcelona between July 2016 and July 2021” (page 8, lines 252-255).
C6. The authors showed the proportion of interested subsets. How about the numbers of the subsets? As clinical information, WBC numbers should be described.
R6. Thanks. We now include this information (page 5, lines 133-134). White blood count (WBC) was quantified at the core biology laboratory of our hospital, following standard, clinical-grade methodology. No differences regarding WBC numbers between stable and progressors patients were observed.
C7. CD56+ T cells might be better than NKT-like cells.
R7. With all due respect, we hold a different opinion regarding this comment. The term "NKT-like" encompasses a broader category of T cells that align with the initial characterization of NKT cells, characterized by the expression of CD3+CD56+. This terminology has gained extensive acceptance in the literature, as demonstrated by several references. For instance, as seen in:
- Jiang Y, Cui X, Cui C, Zhang J, Zhou F, Zhang Z, Fu Y, Xu J, Chu Z, Liu J, Han X, Liao C, Wang Y, Cao Y, Shang H. The function of CD3+CD56+ NKT-like cells in HIV-infected individuals. Biomed Res Int. 2014; doi: 10.1155/2014/863625; and,
- Kaszubowska, L., Foerster, J. & Kmieć, Z. NKT-like (CD3+ CD56+) cells differ from T cells in expression level of cellular protective proteins and sensitivity to stimulation in the process of ageing. Immun Ageing 19, 18 (2022). https://doi.org/10.1186/s12979-022-00274-z).
